# The Relationship between the Soluble Receptor for Advanced Glycation End Products and Oxidative Stress in Patients with Palmoplantar Warts

**DOI:** 10.3390/medicina55100706

**Published:** 2019-10-20

**Authors:** Cristina Iulia Mitran, Ilinca Nicolae, Mircea Tampa, Madalina Irina Mitran, Constantin Caruntu, Maria Isabela Sarbu, Corina Daniela Ene, Clara Matei, Antoniu Cringu Ionescu, Simona Roxana Georgescu, Mircea Ioan Popa

**Affiliations:** 1“Carol Davila” University of Medicine and Pharmacy, 020021 Bucharest, Romania; cristina.iulia.mitran@gmail.com (C.I.M.); madalina.irina.mitran@gmail.com (M.I.M.); isabela_sarbu@yahoo.com (M.I.S.); matei_clara@yahoo.com (C.M.); antoniuginec@yahoo.com (A.C.I.); mircea.ioan.popa@gmail.com (M.I.P.); 2“Cantacuzino” National Medico-Military Institute for Research and Development, 011233 Bucharest, Romania; 3“Victor Babes” Clinical Hospital for Infectious Diseases, 030303 Bucharest, Romania; drnicolaei@yahoo.ro; 4“Prof. N. Paulescu” National Institute of Diabetes, Nutrition and Metabolic Diseases, 011233 Bucharest, Romania; 5“Carol Davila’ Nephrology Hospital”, 010731 Bucharest, Romania; koranik85@yahoo.com

**Keywords:** sRAGE, oxidative stress, inflammation, warts, HPV

## Abstract

*Background and objectives*: Warts are the most common lesions caused by human papillomavirus (HPV). Recent research suggests that oxidative stress and inflammation are involved in the pathogenesis of HPV-related lesions. It has been shown that the soluble receptor for advanced glycation end products (sRAGE) may act as a protective factor against the deleterious effects of inflammation and oxidative stress, two interconnected processes. However, in HPV infection, the role of sRAGE, constitutively expressed in the skin, has not been investigated in previous studies. *Materials and Methods*: In order to analyze the role of sRAGE in warts, we investigated the link between sRAGE and the inflammatory response on one hand, and the relationship between sRAGE and the total oxidant/antioxidant status (TOS/TAS) on the other hand, in both patients with palmoplantar warts (*n* = 24) and healthy subjects as controls (*n* = 28). *Results*: Compared to the control group, our results showed that patients with warts had lower levels of sRAGE (1036.50 ± 207.60 pg/mL vs. 1215.32 ± 266.12 pg/mL, *p* < 0.05), higher serum levels of TOS (3.17 ± 0.27 vs. 2.93 ± 0.22 µmol H2O2 Eq/L, *p* < 0.01), lower serum levels of TAS (1.85 ± 0.12 vs. 2.03 ± 0.14 µmol Trolox Eq/L, *p* < 0.01) and minor variations of the inflammation parameters (high sensitivity-CRP, interleukin-6, fibrinogen, and erythrocyte sedimentation rate). Moreover, in patients with warts, sRAGE positively correlated with TAS (r = 0.43, *p* < 0.05), negatively correlated with TOS (r = −0.90, *p* < 0.01), and there was no significant correlation with inflammation parameters. There were no significant differences regarding the studied parameters between groups when we stratified the patients according to the number of the lesions and disease duration. *Conclusions*: Our results suggest that sRAGE acts as a negative regulator of oxidative stress and could represent a mediator involved in the development of warts. However, we consider that the level of sRAGE cannot be used as a biomarker for the severity of warts. To the best of our knowledge, this is the first study to demonstrate that sRAGE could be involved in HPV pathogenesis and represent a marker of oxidative stress in patients with warts.

## 1. Introduction

Warts are muco-cutaneous lesions caused by human papillomavirus (HPV) [1]. Warts are very common in the general population, with up to one third of the children being affected; the incidence decreases with age. Considering that immunity plays an important role in the development of warts, the incidence increases up to 45% among immunosuppressed individuals [2]. To this date, more than 200 HPV types have been identified [3]. Warts are commonly associated with HPV 1, 2, 4 and 7. In immunosuppressed patients, HPV 75, 76 and 77 were identified. Warts manifest as skin-colored papules with a keratotic surface. Most frequently, the lesions involve the hands and feet, but they can also appear on extension areas (elbows, knees) [4,5]. Transmission of the infection occurs through direct contact in the context of skin microlesions or more rarely, indirectly through contaminated objects [5]. The incubation period ranges from 1 to 20 months; there is no systemic spread of the virus. HPV infects host cells without integrating viral DNA into their genome [6].

Warts occur when HPV infects the upper layers of the skin or mucous membranes, resulting in abnormal and rapid cell growth. There are several conditions that contribute to the acquisition of HPV infection, including incomplete immune system development, CD4+ T-cell dysfunction, other causes of immunosuppression, and the alteration of the normal epithelial barrier [7,8,9,10,11,12,13]. Most often in children, warts resolve spontaneously, without treatment, but the lesions can be persistent in adults. However, recurrences are common in both groups [10,11]. There is no clear evidence of the etiopathogenic mechanisms of HPV infection. Multiple factors contributing to its pathogenesis have been suggested, such as the lack of efficient protective cell mechanisms against the virus and the accumulation of toxic compounds which induce oxidative stress and inflammation [14,15,16,17,18,19]. Oxidative stress and inflammation are two interconnected processes which produce multiple effects on cell function [20]. 

The 45 kDa receptor for advanced glycation end products (RAGE) is a member of the immunoglobulin superfamily, which contains an extracellular region of 320 amino acids (sequence 23–342), a 21-amino acid transmembrane hydrophobic region (sequence 343–363) and a cytoplasmic region of 41 amino acids (sequence 364–404). The extracellular region is composed of two C-type domains (C1 consisting of 124–221 amino acid residues and C2 consisting of 227–317 amino acid residues) and a V-type domain (sequence 23–116 amino acid residues). The V domain confers the ability to bind various ligands. The V and C1 domains are involved in the stability/specificity of the receptor-ligand complex. The C2 domain participates in the dimerization/oligomerization of receptor-ligand complexes. The receptor is anchored to the cell membrane through its transmembrane domain. The endocellular domain is essential for intracellular signaling [21,22,23].

Many studies have shown that RAGE has a large number of ligands, such as advanced glycation end products (AGEs), S100 calgranulin proteins, high motility group box 1 protein, amyloid fibrils, phosphatidylserine, macrophage-1 antigen, lipopolysaccharide from the outer membrane of Gram-negative bacteria, peptidoglycan (present in the majority of bacteria), lipoteichoic acid (a component of many Gram-positive bacteria), bacterial DNA, viral DNA/RNA, yeast cell wall mannans, degraded extracellular matrix components, and modified fibronectin [15,18,24,25].

RAGE has been found during embryonic development, its expression being reduced in adulthood, excepting the lung and skin [17]. Other cells, such as monocytes/macrophages, smooth muscle cells, endothelial cells, fibroblasts, and neuronal cells, do not express physiologically detectable amounts, but receptor expression may be induced during cell stress in various instances including the accumulation of the ligands of RAGE, or when the transcriptional factors which modulate the expression of RAGE are activated [26,27]. Soluble RAGE (sRAGE) acts as a decoy for RAGE ligands, preventing their interactions with membrane RAGE (mRAGE) through a competitive mechanism, and consequently, sRAGE disrupts the generation of oxidative stress and inflammation [14]. Increased amounts of RAGE ligands induce the overexpression of sRAGE [27].

To the best of our knowledge, the role of sRAGE in the onset and progression of warts has not been previously investigated. To analyze the possible molecular mechanisms involved in the development of cutaneous warts, we hypothesized that sRAGE acts as an endogenous factor able to maintain/restore the cutaneous homeostasis. We have analyzed the interaction between circulating sRAGE and inflammatory response on the one hand and the relationship between sRAGE and oxidant/antioxidant status on the other hand in patients with palmoplantar warts (Figure 1).

## 2. Materials and Methods

### 2.1. Study Participants

The patients were selected from those who presented to the Clinic of Dermatology and were diagnosed with palmoplantar warts, the most common HPV-related lesions. All study participants gave their consent to the use of their biological samples in research studies. All the procedures and the experiments performed in the study respect the ethical standards in the Helsinki Declaration, as well as the national law. The study protocol was approved by the Ethics Committee of “Victor Babes Infectious and Tropical Diseases Hospital” (13050/31.07.2017).

Inclusion criteria: otherwise healthy adults, aged 18 years or above; adequate nutritional status; non-smokers; and with no treatment for warts.

Exclusion criteria from the study (conditions widely known as being able to alter, and therefore interfere with parameters of inflammation and oxidative stress): chronic alcohol use, drug abuse; treatment with corticosteroids, immunosuppressant agents and nutritional supplements; and pregnancy and breastfeeding. 

Based on similar demographic characteristics, the study participants were divided into two groups: patients with palmoplantar warts (*n* = 24) and healthy subjects as controls (*n* = 28). We analyzed the serum levels of sRAGE, oxidative stress parameters and markers of inflammation compared to controls. We also evaluated the levels of sRAGE, oxidative stress parameters and markers of inflammation according to the number of the lesions and the duration of the disease. According to the number of the lesions we stratified the patients into three groups: less than 5 lesions (*n* = 11), between 5–10 lesions (*n* = 8) and more than 10 lesions (*n* = 5). The distribution of the patients with warts according to the duration of the disease divided them into three groups: with a history of less than 1 month (*n* = 6), between 1 and 6 months (*n* = 10) and a history longer than 6 months (*n* = 8). 

### 2.2. Laboratory Tests

Biological samples were drawn from the patients and controls enrolled in the study under basal conditions using a holder-vacutainer system. Venous blood collected on anticoagulant (K3EDTA) was used to determine the blood count and erythrocyte sedimentation rate. The samples were processed immediately. The plasma obtained from venous blood collected on heparin was used for serum fibrinogen determination. Serum was obtained from venous blood collected in vacutainer without anticoagulant. The hemolyzed or lactescent samples were rejected. 

sRAGE levels were measured by ELISA method; the sandwich variant and the results were expressed as pg/mL. In the wells of a polystyrene plate in which known antibodies were attached, the unknown antigen solution was added and then incubated. After washing, enzyme-labelled antibodies were added and fixed to the free epitopes of a polyvalent antigen. After incubation, the wells were washed again. The presence of the labelled complex was detected using a chromogenic substrate (BioVision reagents, TECAN analyzer). The absorbance of the resulted yellow product was measured. The intensity of the color of the resulted product is proportional to the amount of sRAGE in the sample. To determine the concentrations of sRAGE in the samples, a standard curve was used. The intensity of the color was measured at 450 nm. 

The following parameters were used to assess oxidative stress: total antioxidant status (TAS), total oxidant status (TOS), and oxidative stress index (OSI).

TOS and TAS levels were determined by spectrophotometric method (Randox reagents, HumaStar 300 analyzer); results were expressed as μmol of H_2_O_2_ equivalent/L serum for TOS and as µmol Trolox equivalent/L serum for TAS. OSI value was calculated using the following formula:(1)$$\mathrm{OSI}\;\left(\mathrm{arbitrary}\;\mathrm{units}\right)=\frac{\mathrm{TOS}\;\left({µ\mathrm{mol}\;H}_{2}O_{2}\mathrm{Eq}/L\right)}{\mathrm{TAS}\;\left(µ\mathrm{mol}\;\mathrm{Trolox}\;\mathrm{Eq}/L\right)}$$

For the early detection of inflammation, the following determinations were used: high sensitivity C-reactive protein (hs-CRP, latex-immunoturbidimetric method), interleukin-6 (IL-6, ELISA method, sandwich variant, automatic reading method), the erythrocyte sedimentation rate (ESR, automatic reading method), and fibrinogen (coagulometric method).

### 2.3. Statistical Analysis

The comparison of obtained experimental data between groups was carried out using t-test. When we compared more than 2 groups we used Kruskal-Wallis test. The relationship between pairs of two parameters was assessed by Spearman’s correlation coefficient after adequate assessment of normality of data using Kolmogorov-Smirnov test. We chose a significance level (*p*) of 0.05 (5%) and a confidence interval of 95% for hypothesis testing.

## 3. Results

The mean serum levels of sRAGE were significantly lower in patients with warts compared to healthy controls (1036.50 ± 207.60 pg/mL vs. 1215.32 ± 266.12 pg/mL, *p* < 0.05) (Table 1). Differences were also obtained for TAS levels (1.85 ± 0.12 vs. 2.03 ± 0.14 µmol Trolox Eq/L, *p* < 0.05), TOS levels (3.17 ± 0.27 vs. 2.93 ± 0.22 µmol H_2_O_2_ Eq/L, *p* < 0.01) and OSI (1.72 ± 0.22 vs. 1.45 ± 0.17, *p* < 0.01) compared to controls, (Table 1). The determination of the markers of inflammation did not reveal a relevant inflammatory process in patients with warts. The only exception was represented by hs-CRP levels. The mean level of hs-CRP was 0.19 ± 0.14 mg/dL in patients with warts and 0.06 ± 0.02 mg/dL in controls (*p* < 0.05). In contrast, IL-6, fibrinogen, and ESR did not show significant differences between the two groups (Table 1).

The serum levels of the studied parameters did not differ significantly according to the number of the lesions between the groups (Table 2).

There were no significantly differences between groups when we stratified patients according to the duration of the disease (Table 3).

In patients with warts, sRAGE levels showed a positive statistically significant association with TAS (rho = 0.43, *p* < 0.05) and a negative statistically significant association with both TOS (rho = −0.90, *p* < 0.01) and OSI (rho = −0.86, *p* < 0.01) (Table 3). There was a lack of correlation between the levels of sRAGE and hs-CRP, IL-6, fibrinogen, and ESR in patients with warts (Table 4).

## 4. Discussion

The serum concentration of sRAGE seems to be modulated by a complex group of factors such as genetic factors, internal and environmental stimuli [27,28,29,30,31,32,33]. It has also been suggested that serum levels of sRAGE are influenced by gender, age, ethnicity, the imbalance between antioxidants and prooxidants and inflammatory processes [27]. There are studies which attribute to sRAGE the role of a potential biomarker of oxidative stress [34]. Some reports have suggested that the overexpression of sRAGE reflects the excessive inflammatory response involved in the progression of endothelial lesions and coagulopathy associated with severe infection. Low levels of sRAGE were associated with increased levels of IL-6, VCAM-1 and PAI-1 as well as with thrombocytopenia [35]. At the same time, in patients with type 1 diabetes, sRAGE overexpression in response to increased levels of AGEs was interpreted as a modality of protection against cell damage. Under these conditions, sRAGE acts as a negative feedback mediator for eliminating AGEs. The overexpression of sRAGE can be considered a weapon against cell damage and a mechanism to regulate the receptor synthesis by modulating the synthesis of enzymes that produce a proteolytic cleavage [36]. 

In our study, we have found that patients with palmoplantar warts had lower serum levels of sRAGE compared to controls. sRAGE downregulation may be a factor involved in HPV pathogenesis; it can be speculated that sRAGE acts as a negative regulator of warts occurrence and could represent an early mediator involved in the onset and development of warts. The decrease in sRAGE levels in patients with palmoplantar warts could be explained by different mechanisms. HPV induces increased proliferation of keratinocytes resulting in a higher rate of glucose metabolism in the infected cells, which stimulates the synthesis of AGEs. Thus, AGEs accumulate in extracellular spaces and interact with sRAGE [37]. Another possible explanation is the disruption of the AGEs-sRAGE axis that might induce a low synthesis of soluble receptors [18,36,38]. sRAGE is cleaved on cell surface through the action of matrix metalloproteinases. The activity of these enzymes is modulated by oxidized lipoproteins [32]. It has also been reported that advanced glycosylation of high density lipoproteins leads to endogenous sRAGE sequestration [32,33]. 

The relationship between RAGE expression in the skin and the level of its ligands remains unclear. In human skin, sRAGE was positively correlated with the expression of genes encoding for ligands of RAGE such as tumor necrosis factor (TNF) alpha, IL-1 alpha, S100B, proapoptotic factors (Fas, Bax), epidermal differentiation markers (involucrin), and proliferating cell nuclear antigen [17]. Another factor that could influence sRAGE activity is the presence of a group of cell surface receptors, AGE-R1, AGE-R2, and AGE-R3, which seem to modulate the endocytosis and degradation of AGEs, thus counteracting the effects of RAGE. AGE-R1 has been shown to reduce oxidative stress induced by AGEs through the inhibition of RAGE signaling pathway [37]. 

Another point analyzed in our study was the investigation of potential mechanisms by which sRAGE is involved in the pathogenesis of warts. To demonstrate this hypothesis, we have performed a complete, simultaneous and comparable analysis of the axis sRAGE – markers of oxidative stress – markers of inflammation in patients with warts and in a control group. First, we have investigated oxidative stress markers (TOS, TAS, OSI) and confirmed the presence of an imbalance between oxidant load and antioxidant defense in patients with warts. Previous studies have suggested that the balance between oxidants and antioxidants plays an important role in the spontaneous regression of HPV infection, and the antioxidant system prevents the effects of oxidative stress and mediates the immune response [39,40]. A recent study has shown that oxidative stress plays an important role in recalcitrant warts [41]. Excessive amounts of oxidants lead to destructive effects, materialized in the structural and functional alteration of lipids, proteins and nucleotides [42]. In this case, the antioxidant systems may become deficient, favoring the perpetuation of oxidative stress. We consider that sRAGE can participate in the restoration of the oxidant/antioxidant balance in patients with warts. This hypothesis is supported by the strong negative correlation between sRAGE and TOS, respectively OSI, and the positive association between sRAGE and TAS. Based on these results, we assign to sRAGE the role of a potential biomarker of oxidative stress in patients with warts. Therefore, the modulation of sRAGE level in HPV patients might influence the progression of the disease. 

In our study, the evaluation of a panel of markers of inflammation did not reveal an inflammatory systemic process in patients with warts. The sRAGE level was not correlated with the levels of the markers of inflammation (IL-6, fibrinogen and ESR) excepting hs-CRP. However, we do not exclude the presence of a proinflammatory environment in infected tissues. The association between sRAGE and hs-CRP has also been proven in several pathological conditions [43]. CRP is synthesized by hepatocytes in response to TNF alpha, IL-1, and IL-6 [44]. The AGEs-RAGE interaction increases the expression of these cytokines; sRAGE and RAGE compete for the same ligands. As a result, low sRAGE levels increase the AGEs-mRAGE interaction, which leads to increased cytokine production. It is known that sRAGE modulates the synthesis of hs-CRP in patients with acute coronary syndrome [45]. 

We have identified that none of the examined parameters was influenced by the extension of warts and disease duration. Our results are in concordance with the study by Sasmaz et al., which has also revealed that markers of oxidative stress (catalase, glucose-6-phosphate dehydrogenase, superoxide dismutase, and malondialdehyde) did not correlate with the duration and the number of the lesions [46]. We consider that the level of sRAGE cannot be used as a biomarker for the severity of warts. The molecular mechanisms by which sRAGE could be involved in the etiopathogenesis of warts are complex and could include the interference between oxidative stress and inflammation.

In our study, we have shown changes of serum levels of sRAGE in patients with palmoplantar warts compared to the control group. Given that warts are produced by HPV we have suggested a possible role of sRAGE in the pathogenesis of HPV infection. Further studies investigating the presence of the virus, its type, and its viral load in the examined patients are needed, in order to establish the exact role of sRAGE in HPV infection. Our findings open new perspectives and pave the way for the investigation of sRAGE in HPV infection.

Currently, there is no data available in the literature on the implication of sRAGE in the deep mechanisms that mediate the appearance and evolution of warts. These findings could help broaden the therapeutic options for HPV lesions. Some studies have shown that sRAGE could be an effective therapeutic target and might be used as a biological agent [21]. It has been suggested that increased concentrations of sRAGE may contribute to the inhibition of the inflammatory signaling pathways [47]. 

## 5. Conclusions

Our study reveals an imbalance between prooxidants and antioxidants in patients with warts. Moreover, we postulate that sRAGE may represent a potential biomarker of oxidative stress in patients with warts. sRAGE acts as a decoy for AGEs, blocking AGEs-RAGE axis and prevent the augmentation of the oxidative processes. The modulation of sRAGE could be a therapeutic alternative or at least an adjuvant treatment in near future. 

## Figures and Tables

**Figure 1 medicina-55-00706-f001:**
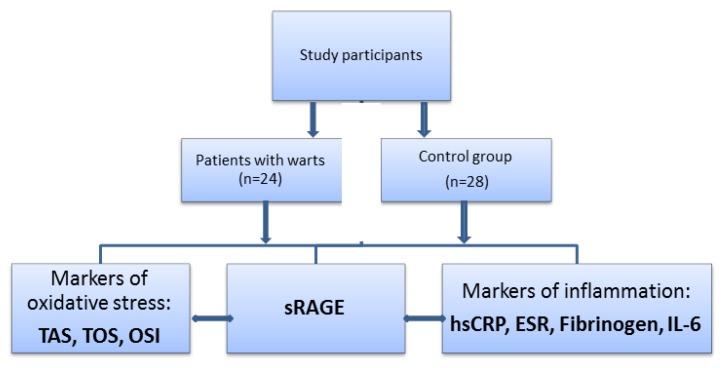
The plan of investigation of sRAGE in association with oxidative stress and inflammation in patients with warts. sRAGE = soluble receptor for advanced glycation end products, TAS = total antioxidant status, TOS = total oxidant status, OSI = oxidative stress index, ESR = erythrocyte sedimentation rate, hs-CRP = high-sensitive C reactive protein, Il-6 = interleukin-6.

**Table 1 medicina-55-00706-t001:** The serum levels of sRAGE, oxidative stress parameters and markers of inflammation in patients with warts versus controls (expressed as mean and standard deviation).

Parameter	Patients with Warts *n* = 24	Controls *n* = 28	*p* Value
sRAGE (pg/mL)	1036.50 ± 207.60	1215.32 ± 266.12	<0.05 *
**Markers of oxidative stress**			
TAS (µmol Trolox Eq/L).	1.85 ± 0.12	2.03 ± 0.14	<0.01 *
TOS (µmol H_2_O_2_ Eq/L)	3.17 ± 0.27	2.93 ± 0.22	<0.01 *
OSI (arbitrary units)	1.72 ± 0.22	1.45 ± 0.17	<0.01 *
**Markers of inflammation**			
hs-CRP (mg/dL)	0.19 ± 0.14	0.06 ± 0.02	<0.01 *
ESR (mm/h)	5.20 ± 3.30	3.80 ± 2.10	>0.05
Fibrinogen (mg/dL)	183.5 ± 59.10	179.6 ± 64.70	>0.05
IL-6 (pg/mL)	7.62 ± 2.60	7.08 ± 2.40	>0.05

*n* = number of the patients. *—statistically significant.

**Table 2 medicina-55-00706-t002:** The serum levels of sRAGE, oxidative stress parameters and markers of inflammation in patients with warts (expressed as mean and standard deviation) according to the number of the lesions.

Parameter	Patients with Warts			*p* Value
	<5 (*n* = 11)	5–10 (*n* = 8)	>10 (*n* = 5)	
sRAGE (pg/mL)	1029.45 ± 237.52	10,562.5 ± 204.47	1020.4 ± 179.90	0.9
**Markers of oxidative stress**				
TAS (µmol Trolox Eq/L).	1.83 ± 0.12	1.85 ± 0.11	1.89 ± 0.17	0.58
TOS (µmol H_2_O_2_ Eq/L)	3.23 ± 0.35	3.10 ± 0.21	3.17 ± 0.24	0.62
OSI (arbitrary units)	1.77 ± 0.23	1.68 ± 0.15	1.70 ± 0.30	0.52
**Markers of inflammation**				
hs-CRP (mg/dL)	0.19 ± 0.16	0.20 ± 0.16	0.19 ± 0.11	0.9
ESR (mm/h)	6.00 ± 3.58	4.50 ± 3.11	4.60 ± 3.13	0.8
Fibrinogen (mg/dL)	171.72 ± 53.20	191.37 ± 57.58	206.4 ± 78.00	0.6
IL-6 (pg/mL)	7.93 ± 2.72	7.49 ± 2.63	7.16 ± 2.93	0.9

The patients were divided into three groups; *n* = number of the patients.

**Table 3 medicina-55-00706-t003:** The serum levels of sRAGE, oxidative stress parameters and markers of inflammation in patients with warts (expressed as mean and standard deviation) according to the duration of the disease (months).

Parameter	Patients with Warts			*p* Value
	<1 (*n* = 6)	1–6 (*n* = 10)	>6 (*n* = 8)	
sRAGE (pg/mL)	1061.00 ± 278.63	1090.50 ± 207.13	950.62 ± 133.72	0.26
**Markers of oxidative stress**				
TAS (µmol Trolox Eq/L).	1.86 ± 0.12	1.88 ± 0.14	1.82 ± 0.12	0.49
TOS (µmol H_2_O_2_ Eq/L)	3.13 ± 0.20	3.16 ± 0.33	3.22 ± 0.30	0.91
OSI (arbitrary units)	1.69 ± 0.10	1.70 ± 0.27	1.78 ± 0.22	0.79
**Markers of inflammation**				
hs-CRP (mg/dL)	0.20 ± 0.13	0.21 ± 0.16	0.19 ± 0.16	0.82
ESR (mm/h)	7.17 ± 3.87	4.00 ± 2.45	5.25 ± 3.41	0.23
Fibrinogen (mg/dL)	182.33 ± 67.45	185.20 ± 55.23	188.25 ± 65.53	0.87
IL-6 (pg/mL)	6.92 ± 3.16	8.12 ± 2.24	7.53 ± 2.89	0.92

The patients were divided into three groups; *n* = number of the patients.

**Table 4 medicina-55-00706-t004:** The relationship between sRAGE and the markers of oxidative stress and inflammation, in patients with warts.

Parameter	Rho	*p* Value
TAS	0.43	<0.05 *
TOS	−0.90	<0.01 *
OSI	−0.86	<0.01 *
hs-CRP	0.11	>0.05
ESR	−0.10	>0.05
Fibrinogen	0.04	>0.05
IL-6	−0.14	>0.05

*—statistically significant.

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
