# Peer review of "The Relationship between the Soluble Receptor for Advanced Glycation End Products and Oxidative Stress in Patients with Palmoplantar Warts"

_medicina, 2019, doi:10.3390/medicina55100706_

Round 1

Reviewer 1 Report

The manuscript entitled “The relationship between the receptor for advanced glycation end products and oxidative stress in patients with palmoplantar warts.” by Dr. Cristina Iulia Mitran is a well written manuscript and results were very clear. The authors demonstrated that the patients with palmoplantar warts had lower serum levels of sRAGE and antioxidant status. The manuscript is really interesting, however, I would like to ask several things because I am not an expert of warts.

The authors mention HPV and warts in introduction part of the manuscript. However, in the section 2.1 (Study participants), subjects were diagnosed with warts but no information about HPV infection. Is HPV the only cause in warts? And I guess HPV have plenty kinds of virus types. Does any relationship between warts and virus type of HPV? Please add sentences to explain background of subjects at this aspect (in the section 2.1).

In conclusion, the authors mentioned sRAGE is a potential biomarker of oxidative stress in patients with warts. However, in my feeling, to detect oxidative stress is much easier than measurement of sRAGE. If the serum levels of sRAGE tend to decrease before onset warts, sRAGE may be a potential biomarker. Does any information about that?

Are difference of prognosis of warts (or severity of warts) related with serum levels of sRAGE?

Author Response

We would like to thank the editors and the reviewers for the careful and thorough review of our manuscript. We have taken into consideration all the suggestions. We consider all the suggestions very useful.

The manuscript entitled “The relationship between the receptor for advanced glycation end products and oxidative stress in patients with palmoplantar warts.” by Dr. Cristina Iulia Mitran is a well written manuscript and results were very clear. The authors demonstrated that the patients with palmoplantar warts had lower serum levels of sRAGE and antioxidant status. The manuscript is really interesting, however, I would like to ask several things because I am not an expert of warts.

The authors mention HPV and warts in introduction part of the manuscript. However, in the section 2.1 (Study participants), subjects were diagnosed with warts but no information about HPV infection. Is HPV the only cause in warts? And I guess HPV have plenty kinds of virus types. Does any relationship between warts and virus type of HPV? Please add sentences to explain background of subjects at this aspect (in the section 2.1).

The only etiological agent of warts is HPV. Warts are commonly associated with HPV 1, 2, 4 and 7. In immunosuppressed patients HPV 75, 76 and 77 were identified.  .

We have added several sentences in the Introduction section regarding the pathogenesis and clinical aspects of warts. We have also introduced data on HPV types involved in the development of warts. In the section 2.2 (Study participants).we have clarified that the patients were diagnosed with warts, caused by HPV.

In conclusion, the authors mentioned sRAGE is a potential biomarker of oxidative stress in patients with warts. However, in my feeling, to detect oxidative stress is much easier than measurement of sRAGE. If the serum levels of sRAGE tend to decrease before onset warts, sRAGE may be a potential biomarker. Does any information about that?

We consider the suggestion very useful. To understand the variation of sRAGE according to the evolution of the disease further studies are needed. To the best of our knowledge, our study is the first study which has investigated sRAGE in patients with warts.

Are difference of prognosis of warts (or seaverity of warts) related with serum levels of sRAGE?

We have stratified the patients according to the duration and the extension of the disease (see Table 2 and Table 3) and there were not significant differences between groups.

Reviewer 2 Report

The manuscript titled, "The relationship between the receptor for advanced glycation end products and oxidative stress in patients with palmoplantar warts", authored by Mitran et al., is an interesting scientific study with clinical relevance. Authors made commendable efforts to present the background, methods and results. 

As stated in the manuscript, this study is the first study reporting the soluble RAGE involvement in HPV pathogenesis. The experimental design is appropriate and the evidence driven interpretations and logical conclusions, make this study suitable to be published in "Medicina". 

I recommend authors to edit the entire manuscript to maintain uniformity in formatting especially spacing

Author Response

We would like to thank the editors and the reviewers for the careful and thorough review of our manuscript. We have taken into consideration all the suggestions. We consider all the suggestions very useful.

The manuscript titled, "The relationship between the receptor for advanced glycation end products and oxidative stress in patients with palmoplantar warts", authored by Mitran et al., is an interesting scientific study with clinical relevance. Authors made commendable efforts to present the background, methods and results. 

As stated in the manuscript, this study is the first study reporting the soluble RAGE involvement in HPV pathogenesis. The experimental design is appropriate and the evidence driven interpretations and logical conclusions, make this study suitable to be published in "Medicina". 

I recommend authors to edit the entire manuscript to maintain uniformity in formatting especially spacing

We have edited the manuscript as suggested by the reviewer.

Reviewer 3 Report

This paper, entitled “The relationship between the receptor for advanced glycation and oxidative stress in patients with palmoplantar warts” investigates the potential role of sRAGE in the pathogenesis of cutaneous lesions related to HPV infection.

The topic is interesting, also considering the lack of literature evidences in this area.

However, in my opinion, some critical points need to be resolved before the paper can be considered for publication.

The main criticisms are as follows:

In the first part of the "introduction" section, clinical and pathogenetic aspects about viral warts should be better described, with appropriate references. The “plan of investigation” illustrated in Figure 1 is unclear. The figure should be revised. The patients included in the study should be stratified on the basis of factors other than those indicated in the inclusion and exclusion criteria, such as: i) the number and extent of the lesions; ii) the duration of the disease; iii) age; iv) comorbidities and drug intake that may affect the immunological status. In the result section, the authors report a statistically significant correlation between sRAGE serum levels, oxidative stress markers, and presence of warts.

It would be necessary to determine whether these differences persist in patients even after warts treatment, or whether in this case, the parameters are in the normal range.

The authors suggest that sRAGE expression may play a role in HPV infection. However, the presence of the virus and its viral load in the examined patients has not been evaluated

Is necessary to improve the English language and to check the spelling; there are few typing errors in the text.

Author Response

We would like to thank the editors and the reviewers for the careful and thorough review of our manuscript. We have taken into consideration all the suggestions. We consider all the suggestions very useful.

This paper, entitled “The relationship between the receptor for advanced glycation and oxidative stress in patients with palmoplantar warts” investigates the potential role of sRAGE in the pathogenesis of cutaneous lesions related to HPV infection.

The topic is interesting, also considering the lack of literature evidences in this area.

However, in my opinion, some critical points need to be resolved before the paper can be considered for publication.

The main criticisms are as follows:

In the first part of the "introduction" section, clinical and pathogenetic aspects about viral warts should be better described, with appropriate references.

Considering the suggestion very useful, we have added sentences in the Introduction section regarding the clinical and pathogenic aspects of warts with appropriate references.

 The “plan of investigation” illustrated in Figure 1 is unclear. The figure should be revised.

We have revised Figure 1.

The patients included in the study should be stratified on the basis of factors other than those indicated in the inclusion and exclusion criteria, such as: i) the number and extent of the lesions; ii) the duration of the disease; iii) age; iv) comorbidities and drug intake that may affect the immunological status. In the result section, the authors report a statistically significant correlation between sRAGE serum levels, oxidative stress markers, and presence of warts.

We have stratified the patients according to the number of the lesions and the duration of the disease (see Table 2 and Table 3).and there were not significant differences between groups. In our study we have included patients with a narrow age range (21-35), with no comorbidities and drug intake that may affect the immunological status and oxidative stress parameters.

It would be necessary to determine whether these differences persist in patients even after warts treatment, or whether in this case, the parameters are in the normal range.

Warts were treated with various therapeutic methods, choosing the best option for each patient, and the different methods used could have a different impact on the measured parameters. However, we consider the suggestion very useful. In a further study we set out to determine the variation of the markers of oxidative stress and inflammation after treatment.

The authors suggest that sRAGE expression may play a role in HPV infection. However, the presence of the virus and its viral load in the examined patients has not been evaluated

In our study we have shown changes of serum levels of sRAGE in patients with palmoplantar warts compared to the control group. Given that warts are produced by HPV we have suggested a possible role of sRAGE in the pathogenesis of HPV infection. Of course, as we have pointed out in the Discussion section, further studies investigating the presence of the virus, its type and its viral load in the examined patients are needed, in order to establish the exact role of sRAGE in HPV infection. Our findings open new perspectives and suggest that further studies on sRAGE are needed, paving the way for the investigation of this biomarker in HPV infection.

We have modified in the article according to the reviewer’s suggestion.

Is necessary to improve the English language and to check the spelling; there are few typing errors in the text.

We have improved the English language and we have checked the spelling throughout the entire article.

Round 2

Reviewer 3 Report

Minor spell checK required